# Optimization Design of Large-Aperture Primary Mirror for a Space Remote Camera

**DOI:** 10.3390/s23125441

**Published:** 2023-06-08

**Authors:** Xiaohan Liu, Kaihui Gu, Meixuan Li, Zhifeng Cheng

**Affiliations:** 1Jilin Engineering Laboratory for Quantum Information Technology, Jilin Engineering Normal University, Changchun 130052, China; lsc20100212@163.com (X.L.); guba510@126.com (K.G.); limx@jlenu.edu.cn (M.L.); 2Changchun Institute of Optics, Fine Mechanics and Physics, Chinese Academy of Sciences, Changchun 130033, China

**Keywords:** primary mirror, optimized design, lightweight structure, finite element simulation (FES), compromise programming method

## Abstract

Lightweight, high stability, and high-temperature adaptability are the primary considerations when designing the primary mirror of a micro/nano satellite remote sensing camera. In this paper, the optimized design and experimental verification of the large-aperture primary mirror of the space camera with a diameter of Φ610 mm is carried out. First, the design performance index of the primary mirror was determined according to the coaxial tri-reflective optical imaging system. Then, SiC, with excellent comprehensive performance, was selected as the primary mirror material. The initial structural parameters of the primary mirror were obtained using the traditional empirical design method. Due to the improvement of SiC material casting complex structure reflector technology level, the initial structure of the primary mirror was improved by integrating the flange with the primary mirror body design. The support force acts directly on the flange, changing the transmission path of the traditional back plate support force, and has the advantage that the primary mirror surface shape accuracy can be maintained for a long time when subjected to shock, vibration, and temperature changes. Then, a parametric optimization algorithm based on the mathematical method of compromise programming was used to optimize the design of the initial structural parameters of the improved primary mirror and the flexible hinge, and finite element simulation was conducted on the optimally designed primary mirror assembly. Simulation results show that the root mean square (RMS) surface error is less than *λ*/50 (*λ* = 632.8 nm) under gravity, 4 °C temperature rise, and 0.01 mm assembly error. The mass of the primary mirror is 8.66 kg. The maximum displacement of the primary mirror assembly is less than 10 μm, and the maximum inclination angle is less than 5″. The fundamental frequency is 203.74 Hz. Finally, after the primary mirror assembly was precision manufactured and assembled, the surface shape accuracy of the primary mirror was tested by ZYGO interferometer, and the test value was 0.02 *λ*. The vibration test of the primary mirror assembly was conducted at a fundamental frequency of 208.25 Hz. This simulation and experimental results show that the optimized design of the primary mirror assembly meets the design requirements of the space camera.

## 1. Introduction

Micro/nano satellite weighing less than 100 kg, which has a high resolution, real-time mobility, and flexibility, is widely used in the fields of environmental monitoring, geological mapping, and meteorological observation [1,2,3]. The satellite has strict control over size and weight, featuring small size, lightweight, and low launch cost. Therefore, there are also strict limitations on the optoelectronic payloads carried on them, the space camera for ground imaging. However, the camera needs to have high stability and good environmental adaptability to ensure clear images during the test, launch, and in-orbit operation. As the core component of the high-resolution space remote sensing camera and the largest optical element, the primary mirror has the role of collecting imaging light, and its surface shape accuracy and structural stability determine the imaging quality of the optical system. Therefore, the design is optimized to be as light as possible while ensuring high stability and high accuracy of the primary mirror assembly [4,5,6]. The primary mirror material, lightweight structure type, and support method will directly affect the reflector surface shape accuracy, stability, and lightweight. High stability and lightweight are contradictory to each other. The larger the aperture of the primary mirror, the more difficult it is to design, process, and manufacture. Therefore, how to ensure that the large aperture primary mirror (diameter usually greater than 500 mm) has high stability, high surface shape accuracy, and ultra-lightweight structure, and when the environment changes, the primary mirror accuracy change can be maintained within the allowed range for a long time, becomes one of the technical difficulties in the optimal design and processing of space cameras.

In the early days, the primary mirror design was mainly based on experience. However, now there are more and more optimization design methods, mainly optical system integration size optimization, topology optimization, compromise programming parametric design, etc., and computer simulation analysis, which greatly improves the optimization design efficiency and accuracy. Scholars at home and abroad have conducted much in-depth research on the lightweight form, support method, and optimization method of the primary mirror of a space camera. Meijun Zhang et al. studied the primary mirror with a fan-shaped lightweight unit and 3-point support structure at the back and designed the primary mirror and flexible hinge structure parameters using the optical-mechanical integration dimensional optimization method to obtain the primary mirror assembly with wide temperature adaptability [7]. Yanjun Qu et al. compared and analyzed the effect of triangular and hexagonal weight reduction grooves on the back of the primary mirror on the lightweight rate and the optical performance of the mirror and designed a 610 mm diameter primary mirror assembly using a peripheral flexible support method. Its surface shape accuracy reached 3.58 nm under gravity conditions [8]. Fengchang Liu et al. studied the primary mirror structure with 6-point back support and triangular lightweight form and optimized the design of the primary mirror structure parameters using a compromise programming method and topology optimization algorithm, which improved the overall performance of the primary mirror and reduced the weight, RMSx, and RMSz by 8.5%, 14.3%, and 10.5% in turn [9]. Qin Tao et al. used a hybrid IPSO-IAGA-BPNN algorithm to optimize the structural parameters of the reflector, and the advantages and disadvantages of different optimization methods are also analyzed [10]. Zongxuan Li et al. studied the silicon carbide primary mirror of a space telescope with the semi-enclosed triangular lightweight form at the back and three-point support method at the back, optimized the design and fabricated the primary mirror with a diameter Φ760 mm and flexible hinge, and the surface shape accuracy reached 0.02 *λ* [11]. Hagyong Kihm et al. used a multi-objective genetic algorithm to optimize the ZERODUR^®^ primary mirror and a new bipod flexure, which met the design objectives [12]. Shutian Liu et al. used the topology optimization method to design a large-aperture primary mirror structure of a space telescope with a lightweight hexagonal form and 9-point back support [13]. In order to design the primary mirror structure more accurately and optimally, it has become a new research direction to use multiple methods to integrate the optimal design. The primary mirror lightweight unit is generally triangular, sector, circular, square, square hexagonal, etc. The triangular lightweight form of the mirror has the best stiffness, and the sector has the highest lightweight rate. In order to obtain a high lightweight rate and good optical performance, two or more lightweight forms are combined to make up for the disadvantages of a single structure. The primary mirror support methods are center support, back support, and circumferential support. Center support is generally used for small aperture primary mirrors. Back support is commonly used for medium aperture mirrors and above, while large aperture primary mirrors often use a combination of back support and circumferential support type [14]. According to different support methods to choose a different lightweight form, the peripheral support is mostly used in hexagonal and fan-shaped lightweight forms, while the back support is often used in triangular lightweight forms.

Based on fully studying the existing technology of the space camera primary mirror and optimizing the design theory and method, the integrated casting structure of the flange and mirror body was adopted, and the external force was applied on the flange through the flexible hinge, which avoids the decrease of mirror surface shape accuracy caused by the external force acting directly on the mirror body when using back multi-point support and peripheral support, and effectively reduces the sensitivity of mirror surface shape accuracy to external environment changes. According to the manufacturing process, the primary mirror lightweight form is a combination of triangular shape with high stiffness and sector shape with a high lightweight ratio. A parametric optimization design method based on the compromise programming theory was used to establish an optimized mathematical model of the Φ610 mm aperture SiC primary mirror after a multi-objective optimized design, the initial structural parameters of the flexible hinge and the mirror were carried out. The effect of the dimensional parameters of the mirror assembly is analyzed with the first intrinsic frequency, the surface shape error (RMS) under various operating conditions, and the mass as constraints by using the multi-target comprehensive performance as the objective function. The finite element simulation of the primary mirror assembly was conducted, and the simulated value of the root mean square (RMS) value of the mirror was much smaller than the design requirement, and the fundamental frequency of the mirror component was higher than the index value. The mass of the primary mirror after the parameter optimization design has a substantial decrease compared with the initial structural mass, which is less than the maximum mass required by the design. Finally, the surface shape error interference test and mechanical vibration test were conducted on the primary mirror assembly, and the test results all met the engineering requirements.

## 2. Optical System Design

To realize the lightweight and miniaturization of the satellite, the space camera adopts a coaxial triple reflection optical system and folds the optical path through multiple folding mirrors to reduce the overall size of the space optical remote sensor, making the whole camera small in size and light in weight. Due to the smaller detector pixel size, the size of the whole camera will also be reduced, which is conducive to achieving the total overall satellite index. In this paper, a large surface array CMOS detector with 2.8 um image elements is used as the imaging detector of the camera.

### 2.1. Focal Length of the Optical System

From the perspective of geometrical optics, there is the following relationship between the resolution of ground pixels GSD and focal length $f^{\prime }$, detector pixel size α, and flight height H, as shown in Equation (1):(1)$$GSD=\frac{\alpha \times H}{f^{\prime }}$$

The focal length $f^{\prime }$ can be calculated as follows:(2)$$f^{\prime }=\frac{\alpha \times H}{GSD}$$

In Formula (2), α is the pixel size of the CMOS detector with a large area array, α=2.8 μm, H is flying height, H=500 km, the GSD is the resolution of ground pixels, GSD=0.5 m and the focal length $f^{\prime }=2.8\;m$.

### 2.2. Diameter of the Primary Mirror

The resolution of ground pixels GSD, the effective aperture of the optical system D, which determines the size of the primary mirror diameter, the solar synchronous orbit altitude H, and the central wavelength satisfy Equation (3).
(3)$$D\geq \frac{H\lambda }{GSD}$$

According to the index requirements, the center wavelength λ should be taken at 600 nm, and the optical aperture of the camera should be Φ600 mm, so the size of the mirror diameter is determined to be Φ610 mm. The imaging system first converges the light on the secondary mirror with the largest size optical element primary mirror; then, the secondary mirror reflects the imaging light to the first folding mirror. The folding mirror reflects the light to the third reflector, and the light passes through the second folding mirror and converges to the CMOS detector for imaging. Usually, the light bar is set at the first image plane, which can eliminate more than 90% of stray light and is conducive to improving the imaging quality. The coaxial triple reflection optical system is displayed in Figure 1.

The MTF of the coaxial triple reflection optical system at the Nyquist frequency is shown in Figure 2. The design transfer function curve of the optical system for reaching the diffraction limit is given, and the average modulation transfer function at the Nyquist frequency (178 PL/mm) reaches 0.245.

The primary mirror studied in this paper is a large aperture mirror in a space remote sensing camera, its mirror surface is aspheric, and its expression is as follows:(4)$$\left\{\begin{array}{l} s(r)=\frac{Cr^{2}}{1+\sqrt{1-(1+k)C^{2}r^{2}}}\\ C=1/R\\ r^{2}=x^{2}+y^{2} \end{array}\right.$$

In Formula (4), C is surface curvature. R is the radius of curvature of the base circle, R=−938.12 mm. r is Polar coordinates. k is the quadratic surface coefficient k=−0.9768. The traditional optimal design for large aperture mirrors is the empirical design method. The material, lightweight form, and supporting mode of mirrors mainly depend on the aperture of the mirror, design index constraints and manufacturing technology, etc. The initial structural dimensions of mirrors are obtained by referring to various empirical formulas of the structural parameters of the mirror.

### 2.3. Design Requirements

When the space camera works in ground test, launch, and in-orbit operation, the primary mirror must have the ability to keep a high surface shape accuracy constant for a long time, so the mirror assembly must have high stability, high stiffness, and high reliability, and the mass should not be larger than the maximum limit value. A rectangular coordinate system is established in the space camera optical system; the optical axis direction is defined as Z-axis, the direction from the primary mirror to the secondary mirror is positive, and the rear surface of the primary mirror is specified as the XOY plane. According to the technical index requirements assigned by the optical system design, in terms of the static aspect, the face shape error of the primary mirror should meet PV ≤ 63.3 nm and RMS ≤ 12.6 nm under the self-weight deformation in X, Y, and *Z* directions, 4° temperature rise and 0.01 mm assembly error. Since the space camera is installed with a strict temperature control device that controls the camera to vary within a temperature range of ±4°, it is only necessary to consider the deformation amount for a 4° temperature rise of the mirror. In addition, the mirror will produce certain errors during processing and assembly. With a maximum flatness error of 0.01 mm, analyze whether the deformation of the primary mirror is less than the requirement of the design index. In terms of dynamics, to obtain high stability of the component structure, its fundamental frequency cannot be lower than 150 Hz. Due to launch cost constraints, the mirror is not more than 10 kg. According to the requirements of the error allocation index of the optical system of the space camera, the detailed design index of the primary mirror assembly is listed in Table 1.

## 3. Design of Initial Parameters of the Primary Mirror

### 3.1. Primary Mirror Materials

The materials used for space camera mirrors are ZERODUR^®^, silicon carbide, monocrystalline silicon, beryllium metal, aluminum alloy, etc. ZERODUR^®^ has a low coefficient of linear expansion and can be machined for weight reduction, but the material is brittle and easy to collapse during processing, and the specific stiffness is low and easily deformed by an external force. Monocrystalline silicon is commonly used in infrared systems, and the diameter of the reflector is generally less than Φ200 mm due to the limitation of the process. Beryllium metal material has high specific stiffness, low density, and good machinability and is the ideal material for space reflectors, but the material beryllium metal is very toxic and seriously endangers the safety of processors, so it is generally not used. Aluminum alloy material is inexpensive and commonly used in infrared systems. However, the denseness of the material is not enough, and the aluminum alloy for the mirror needs to be imported from Holland, and the diameter of the aluminum alloy mirror is generally less than Φ300 mm. Silicon carbide material has the advantages of high specific stiffness, low coefficient of linear expansion, good performance stability, and high precision of surface shape; its comprehensive performance is optimal and has unique advantages in mechanical and thermophysical properties, which are widely used in space cameras. Changchun Institute of Optics, Fine Mechanics, and Physics has mature and advanced manufacturing and processing technologies in SiC mirror casting, precision machining, polishing, and environmental testing [15]. Therefore, considering many factors such as material performance, processing technology, and processing cost. In this paper, SiC material is chosen as the primary mirror material. When the surrounding temperature changes, thermal stress is generated due to the coefficient of thermal expansion of the materials being different, resulting in a decrease in the accuracy of the surface shape of the primary mirror. The flexible support structure should use the linear expansion coefficient adjustable Inva (4J32) by adjusting its composition ratio so that the thermal expansion coefficient and SiC are consistent, as far as possible, to eliminate the impact of thermal stress. The material characteristics are listed in Table 2.

### 3.2. Initial Structural Design of the Primary Mirror

In the structure design of a reflector, the mirror aperture, thickness, modulus of elasticity, and other factors are usually considered. According to Robert [16] and others, who studied empirical formulas for parameters such as self-weight deformation and diameter-thickness ratio, the thickness of the primary mirror can be calculated. The empirical formula is shown as follows:(5)$$\delta_{\max}=\frac{3\rho gr^{4}}{16Et^{2}}=\frac{3\rho g{\left(D/t\right)}^{2}D^{2}}{256E}$$

In Equation (5), $\delta_{\max}$ is the largest distortion of the primary reflector. E is the modulus of elasticity of the silicon carbide material. ρ is material density. t is the thickness and r is radius. According to the calculated value of Equation (5) and the optical design results, the initial thickness of the reflector was set at 80 mm.

There are usually four types of support for the primary mirror, that is, central support, side support, back support, and composite support. The number of support holes is usually a multiple of three and symmetrically distributed about the central axis. When using traditional back support, the sleeve is usually glued to the back support holes of the primary mirror, and the flexure is then screwed to the sleeve. When subjected to additional external forces, the flexible joints counteract the stresses to reduce the impact on the surface accuracy, and large-diameter mirrors are usually back-supported. The empirical formula for the minimum number of support points required for the back support of a circular mirror studied by Hallet et al. is shown below [17]:(6)$$N=\frac{1.5r^{2}}{t}\sqrt{\frac{\rho g}{E\delta }}$$

From Equation (6), the support point N is calculated to be 3.

According to the experience, the diameter of the circle where the support points are located is $\sqrt{3}/3$ times the diameter of the reflector to obtain the optimal surface shape accuracy during processing. The calculation determined that the support points are symmetrically distributed on a circle with a diameter of Φ350 mm.

The form of the SiC reflector lightweight cell directly affects the lightweight rate, stiffness, and processing difficulty of the reflector. To ensure that the mirror has good stability and surface accuracy with dead weight and temperature change, lightweight cells should be considered in the initial structural design. The common lightweight structures are mainly triangular, hexagonal, fan-shaped, and combination forms [18,19,20,21,22]. The stiffness of the sector brace is low, and the stress transfer effect is not good, but the lightweight rate is higher. The triangular-shaped structural slot provides better stiffness, stability, and higher optical quality than other weight-reducing structures. Lightweight mirror back is usually divided into three types: open, closed, and semi-open [23]. Considering the strength of the primary mirror, the lightweight rate, and the manufacturing difficulty of the mirror blank, the back of the primary mirror adopts a triangular structure lightweight unit, which can effectively improve the specific stiffness and lightweight rate of the mirror body. The fan-shaped area near the center axis of the primary mirror adopts a semi-open structure because the triangular structure of the lightweight unit will increase the difficulty of processing the mirror body, so the lightweight unit is mainly triangular, and the fan-shaped unit is used in the center. The primary reflector mirror is concave, and the edge of the primary mirror is easy to be chipped during processing, leaving a margin of 5 mm on one side, so the final mirror diameter is Φ610 mm. The thickness of the mirror body is 5 mm, the diameter of the holes is Φ50 mm, the thickness of the holes is 5.5 mm, and the width of the other ribs is 3 mm, and the initial structure of the primary mirror based on the empirical formula is shown in Figure 3.

### 3.3. Improvement Structure Based on the Manufacturing Process of the Mirror Blank

The casting of primary mirror blanks using reactive sintered silicon carbide (RB-SiC) material. The shrinkage of this process is only 1–2%, and the size of the primary mirror can be well controlled. Therefore the mirror blank structure with a high lightweight, complex shape, and large aperture can be prepared by this process. The initial primary mirror adopted three-point back support. There is a problem in that the force direction and force transmission path are consistent. When the primary mirror body is affected by microgravity and temperature changes, the force direction and force transmission path are consistent, which leads to a decrease in mirror surface accuracy. In addition, with the development of silicon carbide mirror manufacture and processing technology, silicon carbide mirror with higher precision and lightweight rate and complex structure blank have been applied to space cameras which promotes the development of lightweight camera technology. The current ability to cast complex structures of silicon carbide mirror blanks provides greater design flexibility to obtain highly lightweight structures and high face shape accuracy for space mirrors. After considering the manufacturing process of mirror blank and the difficulty of mirror assembly and adjustment, the back three points act on the flange surface integrated with the mirror body instead of the initial back three points support. When stress and strain are produced by the change of self-weight and temperature, the mirror surface is not directly damaged by internal stress which acts on the flange surface, the deformation is counteracted by the flexible hinge installed on the flange, and thus long-term stability of mirror surface precision is ensured. The flange surface provides a processing benchmark for the precision manufacturing of the mirror surface, which is convenient for processing and manufacturing and is convenient for assembly and adjustment, and that avoids the complex assembly of the back support of the mirror body and short lead time. The weight of the initial structure of the primary mirror is 13.802 kg. The initial structure of the primary mirror after process improvement is as follows in Figure 4.

## 4. Optimization Design of the Primary Mirror

### 4.1. Optimization Mathematical Model Based on Compromise Programming Theory

In many cases, the sub-goals often conflict with each other, so it is impossible to optimize all the sub-goals to the best value. The solution is to compromise among multiple goals and make each sub-goal reach the best as much as possible. Parametric design is a multi-objective optimization process that needs to convert multi-objective optimization into single-objective optimization. The compromise programming theory is conducted to set the optimization function. Therefore, all multi-objective optimization mathematical models can all be represented by the following equation [24,25,26]:(7)$$\left\{\begin{array}{l} find\;x=\left(x_{1},x_{2},x_{3},\cdots ,x_{m}\right)\\ \min f_{i}\left(x\right),i=1,2,3,\cdots ,n\\ s.t.\;x\in X \end{array}\right.$$

In Formula (7), m is the number of variables. n is the total number of objective functions. x is an optimization variable. $f_{i}\left(x\right)$ is sub-target functions. X is constraint sets of optimized variables.

Compare the goals of different measurement methods and transform them into a single-goal optimization Formula (8).
(8)$$\min F\left(t\right)={\left\{\sum_{k=1}^{n}\mu_{k}^{p}{\left[\frac{f_{k}^{\left(t\right)}-Q_{k}^{\min}}{Q_{k}^{\max}-Q_{k}^{\min}}\right]}^{p}\right\}}^{\frac{1}{p}}$$

In Formula (8), $Q_{k}^{\max}$ and $Q_{k}^{\min}$ are the maximal and minimal values of the sub-target in order. $\mu_{k}^{p}$ is the ratio of the weights of the *k*th target. p is the index of distance.

By optimizing the parameters of the primary mirror assembly of the space camera, the surface accuracy and fundamental frequency of the primary mirror can meet the engineering requirements. The mass is also less than the maximum limited index. The shape accuracy, fundamental frequency, and quality are contradictory. The smaller quality of the reflector, the smaller the first-order natural frequency, and conversely, the larger the surface shape error(RMS). Different measurement methods for fundamental frequency, mass, and surface shape errors, so it is suitable to turn the contradictory multi-objective into a single objective by compromise programming mathematics method and optimize the final compromise solution. Using the compromise programming theory, the mathematic model of the parametric compromise programming is shown in (9):(9)$$\min F\left(t\right)={\left\{\begin{array}{l} \mu_{x}^{2}{\left[\frac{R_{x}\left(t\right)-R_{x}^{\min}}{R_{x}^{\max}-R_{x}^{\min}}\right]}^{2}+\mu_{y}^{2}{\left[\frac{R_{y}\left(t\right)-R_{y}^{\min}}{R_{y}^{\max}-R_{y}^{\min}}\right]}^{2}+\mu_{z}^{2}{\left[\frac{R_{z}\left(t\right)-R_{z}^{\min}}{R_{z}^{\max}-R_{z}^{\min}}\right]}^{2}+\\ \mu_{m}^{2}{\left[\frac{m\left(t\right)-m^{\min}}{m^{\max}-m^{\min}}\right]}^{2}+\mu_{f}^{2}{\left[\frac{f\left(t\right)-f^{\min}}{f^{\max}-f^{\min}}\right]}^{2}+\mu_{T}^{2}{\left[\frac{R_{T}\left(t\right)-R_{T}^{\min}}{R_{T}^{\max}-R_{T}^{\min}}\right]}^{2} \end{array}\right\}}^{\frac{1}{2}}$$

In Formula (9), $R_{x}^{\left(t\right)}$ is the RMS under the condition of gravity in the x-direction. $R_{y}^{\left(t\right)}$ is the RMS under the condition of gravity in the Y-direction. $R_{z}^{\left(t\right)}$ is the RMS of the primary mirror under the condition of gravity in the Z-direction. $R^{\min}$ and $R^{\max}$ are the minimum and maximum values at different working conditions, respectively. $m^{\max}$ and $m^{\min}$ are the maximum and minimum values of the quality. $m\left(t\right)$ is the quality of the reflector. $f\left(t\right)$ is the fundamental frequency. $f^{\max}$ and $f^{\min}$ are maximal and minimal values of frequency. μ is the power of each optimized sub-objective. $R_{T}\left(t\right)$ is the mirror-shaped error caused by the temperature change.

### 4.2. Design of the Primary Mirror and Flexible Hinge

In the traditional design of a SiC mirror of a space camera, considering the manufacturability of SiC mirror blank, the thickness and height of the back support rib are usually designed according to consistency, and the values are discrete, which greatly reduces the design difficulty. However, this design method can not make the mirror lightweight to the extreme, and there are some design defects. The transmission of the supporting force on the back of the mirror is mainly determined by the structural type, thickness, and height of the lightweight ribs. If the height and width of the lightweight ribs increase, the mass of the non-important areas will increase, which will lead to deterioration of the surface accuracy and the increase of the dead weight deformation. In this paper, the lightweight holes in different areas of the primary mirror were grouped according to certain rules. Different groups had different rib thicknesses and rib heights, so flanging was designed in important areas under stress. By adjusting and controlling the design variables of each group of parameters, the degree of freedom of design was increased, and the surface accuracy of the primary mirror was gradually improved. Because of the centrosymmetric structure of the mirrors, the lightweight bars can be grouped according to the distance between each group of bars and the center of the circle; that is, the geometric position and lightweight bars can also be grouped according to the force transmission path under specific working conditions. To obtain the comprehensive performance of the primary mirror assembly, such as high surface precision, high stiffness and high structural stability, the primary mirror and flexible hinge were optimized together. The optimization parameters of the primary mirror and the flexible hingle are shown in Figure 5. Among them, Figure 5a shows the optimized parameters of the primary mirror, ①–⑥ are the rib width, ⑦ is the diameter of the circle where the support point is located, ⑨ is the wall thickness of the primary mirror body, and ⑧ and ⑩ are the rib height. Figure 5b shows the optimized parameters of the flexure hinge, L1 and L4 are the slot widths, L2 and L5 are the distances between parallel slot centers, and L3 and L7 are the distances between the slots on the same circle. L6 is the distance between the slot center and the end face. The flexible hingle was a connection part between the primary mirror and the back plate, which was a key component. Due to the inconsistent coefficient of linear expansion, flexible grooves are used mainly to reduce the effects of thermal and simultaneous assembly stresses between the mirror and the backplane.

The structure of lightweight ribs and the concave groove size of flexible hinges were selected as the initial structure of the optimization design. The optimized design variables included mirror thickness, flanging size, lightweight rib thickness, rib height, the mirror thickness, groove size of flexible hinge and position, etc. In the optimization process, the objective function was the minimum value of the mathematical model of the compromise program, and the mirror surface precision (RMS) and resonance frequency under gravity conditions in three directions were constraint functions. In addition, the factors such as mass minimization design and processing technology level of the SiC mirror should be considered comprehensively, and the constraints in Table 1 should be satisfied at the same time. Because the primary mirror assembly was processed, assembled, and adjusted in a constant temperature room of 20 °C when the space camera was working in orbit, there is a temperature control device to keep the camera in the range of 20 ± 4 °C all the time, so the temperature rise of 4 °C is the constraint condition of temperature environment. To ensure the minimum quality and minimize the thickness of lightweight ribs. Considering the processing and manufacturing capability of the SiC reflector, the minimum thickness of the tendons is 2 mm. The optimizing formulas of the primary mirror assembly can be defined, and the optimization formula for the main mirror component is defined, and the target functions and restraints are given in Equation (10).
(10)$$\begin{array}{l} find\;{\left(t_{i},\;L_{j}\right)}^{T}\;\mathrm{for}\;i=1,\;2,\;3,\;\cdots ,10;\;j=1,\;2,\;3,\;\cdots ,7\\ \min F(t),\;F\left(t\right)={\left\{\sum_{k=1}^{n}\mu_{k}^{p}{\left[\frac{f_{k}^{\left(t\right)}-Q_{k}^{\min}}{Q_{k}^{\max}-Q_{k}^{\min}}\right]}^{p}\right\}}^{\frac{1}{p}}\\ s.t.\left\{\begin{array}{l} \mathrm{PV}\leq \frac{\lambda }{10}=62.3,\;\lambda =632.8\;nm\\ RMS\leq \frac{\lambda }{50}=12.6,\;\lambda =632.8\;nm\\ mass\leq M_{0}=5\;\mathrm{kg}\\ f_{1}\geq f_{0}=150\;\mathrm{Hz}\\ 20\;{}^{\circ}C\leq T\leq 24\;{}^{\circ}C\\ t_{ilow}\leq \left(t_{i}\right)\leq t_{iup}\\ L_{jlow}\leq \left(L_{j}\right)\leq L_{jup}\\ \; \end{array}\right. \end{array}$$

Parametric optimization design is a design process by modifying the initial thickness and height of lightweight bars and calculating the engineering results by the computer, which realizes the automation of the design process. $F\left(t\right)$ is the $\min F\left(t\right)$ of Formula (9). RMS is the average surface error. PV is the maximum error of the reflector under the dead weight conditions in three directions (X, Y, Z). $M_{0}$ is the maximum quality. Mass is the actual mass of the reflector. $f_{1}$ is first-order frequency. $f_{0}$ is the lowest basic frequency ($f_{0}$ is greater than 150 Hz). $t_{i}$ is the design variable. $t_{\mathrm{ilow}}$ is the minimum value. $t_{iup}$ is the maximum value. $t_{i}$ is rib thickness, flanging size, mirror body thickness, rib height, etc. $L_{j}$ is the optimization variable of flexible hinge which refers to the slot size and position size in three directions. $t_{\mathrm{ilow}}$ and $t_{iup}$ are the lowest and highest thickness limit values in order. $L_{\mathrm{jlow}}$ and $L_{\mathrm{jup}}$ are the smallest and the largest width limit value in turn.

### 4.3. Optimization Results

The initial structure was optimally designed according to Equation (10), and the original parameters and optimal results of the primary mirror and flexure hinge are shown in Table 3. The optimized primary mirror assembly and flexible hinge structure are shown in Figure 6. In the optimization design process, the surface accuracy due to temperature change, gravity, and 0.01 mm assembly error under working conditions is minimized as the optimization target function. The weight and first-order frequency are used as constraints. The parameters of the flexible hinge include lamella thickness, slot width, slot length, and inter-slot distance. Therefore, the optimized design of dimensional parameters with a mathematical model based on a compromise solution is used to consider multiple conflicting objectives and improve the optimization efficiency. Therefore, the compromise programming method is of reference value for the optimal design of the primary mirror.

It can be seen from the results that the thickness and height of reinforcement and the diameter of the circle where the three supporting positions were located have a big influence on the shape accuracy of the primary mirror. The flexible slot is mainly used to reduce the effect of thermal stress, and the slot widths L3 and L4 have a greater effect on the fundamental frequency of the reflector assembly. It gradually became shorter from the center of the circle to the edge of the mirror, and the rib height was the highest at the center of the circle. It can be concluded that the mirror force was transmitted from the edge to the center, so the force near the center of the circle was the largest. For this place, the semi-closed structure design was carried out to strengthen the local stiffness to improve the surface accuracy. To reduce the mass of the mirror, a lightweight design of the ribs can be carried out. After the multi-objective optimization design, the optimized value of rib ⑥ and thickness ⑨ reached their lower bounds, but rib ① reached upper bounds. The rib ⑤ is reduced from the original 10 mm to the optimized 8 mm. At the same time, dimensions L3, rib ③, and rib ① were increased. On the contrary, the others were diminished. The quality of the mirror is reduced from the initial 13.802 kg to the optimized 8.66 kg, which is less than 10 kg mentioned in Table 1.

## 5. Engineering Analysis

In the modern engineering design process, finite element analysis (FEA) is widely used in the optimal design of optical machine structures to facilitate early detection of design defects such as substandard structural performance and can easily perform static and dynamic simulation analysis of virtual prototypes. Automated meshing is used to build a meshed model of the primary mirror assembly, including the primary mirror, the flexible hinge, and the back plate, as shown in Figure 7, with a total cell count of 81,870,048 and nodes of 42,570,905. Finally, we applied loads and constraints to the primary mirror assembly and performed static analysis on the face shape accuracy under the working conditions of gravity load in three directions (X, Y, and Z), temperature load, and assembly error of the primary mirror assembly, and performed dynamics simulation analysis such as modal analysis and frequency response analysis on the primary mirror assembly.

### 5.1. Static Analysis

A static analysis of the primary mirror assembly simulation model was performed, including deformation analysis under 4 °C of temperature rise, 0.01 mm of assembly error, and self-weight working conditions in different directions. When the screw hole at the bottom of the primary mirror back plate is a fixed constraint, the primary mirror assembly is simulated for deformation under self-weight conditions after applying gravity loads in X, Y, and Z directions, respectively. To ensure the uniform free expansion of the primary mirror assembly under a 4 °C temperature rise load, the assembly was constrained at three nodes, one of which was fully constrained, and the other two nodes were semi-constrained and were able to perform free expansion to simulate the free state. The reflector assembly was set up at a starting temperature of 20 °C, and a temperature rise load of 4 °C was applied for thermodynamic simulation analysis. The design was usually done separately for individual components such as the primary mirror, flexible hinge, or back plate. Three parts were connected through the screw holes. The structural optimization design and simulation analysis was a coupled design of the three components. To obtain more accurate optimization results, the three components were designed and simulated together.

To prove the static characteristics of the primary mirror assembly, the assembly was subjected to a gravitational load of 1 g, a 4 °C temperature load, and an assembly error of 0.01 mm at the support points in the Y and Z directions in turn. For the assembly error analysis of the primary mirror assembly model, finite element simulation analysis was performed after setting a forced displacement of 0.01 mm on the connection end plane, and the analyzed primary mirror surface shape data were input into the facet analysis program for fitting to obtain the surface shape accuracy value of the mirror when the assembly error was 0.01 mm, and the impact of the assembly error on the surface shape accuracy was analyzed to guide the assembly of the primary mirror assembly optical machine system. The optimized surface shape error cloud of the primary mirror is shown in Figure 8. When the gravity load of 1 g in X, Y, and Z directions was applied separately, the root mean square error of the mirror surface was 4.803 nm, 4.782 nm, and 5.941 nm, respectively, which meets the design requirement of 12.6 nm. The maximum displacement is 4.854 μm, which is less than the index of 10 μm. The maximum inclination angle around X or Y is 2.860″, which is less than 5″. The RMS value of the primary mirror is 0.553 nm under 4° temperature rise conditions, and the above analysis results meet the design requirements shown in Table 1. The finite element analysis results of the primary assembly are summarized in Table 4.

### 5.2. Dynamic Analysis

To ensure that the space camera can not only work normally in space and maintain sufficient accuracy but also not be damaged and not produce residual deformation during launching and carrying, it is required that the camera should have sufficient rigidity and strength. At the same time, the space camera should have good dimensional stability under the harsh temperature environment in space to ensure that the changes of the visual axis and mirror surface shape error are within an allowable range. Therefore, dynamic engineering analysis should be carried out on optical-mechanical structures to check the feasibility of the design scheme and provide a scientific basis for selecting reasonable design parameters. To ensure the imaging quality of the camera, it is required that the support structure of a space camera needs to have good dynamic stiffness. Therefore, the camera will not shake under external mechanical disturbance when working [27]. The dynamic stiffness of the camera structure depends on the dynamic characteristics, and it is chiefly measured by the mode shape and natural frequency of the structure. The larger the natural frequency, especially the first-order frequency, the higher the dynamic stiffness of the structure and the better the specific stiffness of the overall structure. The corresponding lower vibration modes should be avoided from occurring in structural parts connected to optical elements directly.

#### 5.2.1. Modal Analysis

The primary mirror assembly suffers damage during a ground test, transportation, launch, and in-orbit operation due to impact, vibration, and overload. The reflector assembly should have high enough stiffness and strength with good dynamic characteristics to avoid damage. The back plate screw holes in the reflector assembly are fixed and restrained, and the modal analysis was carried out.

The results of the first six natural modes of vibration analysis are presented in Figure 9. The first-order frequency of the reflector assembly is 203.74 Hz, which is better than the design requirement of 150 Hz. The vibration mode of the primary mirror assembly oscillates around the *X*-axis, indicating that the primary mirror assembly has a high enough dynamic stiffness to resist the damage caused by resonance. The first six frequencies and vibration modes are displayed in Table 5.

#### 5.2.2. Frequency Response Analysis

The frequency response analysis is used to determine the stationary response of the structure under sinusoidal excitation and the dynamic response of the primary mirror assembly structure under each frequency of periodic oscillation. According to the overall design requirements of the micro/nano satellite, the response of the primary mirror assembly was analyzed under the low-frequency sinusoidal sweep vibration conditions from 0 to 300 Hz. The back plate of the mirror and the vibration test bench were loaded with a forced motion load displacement of 1 mm and a structural damping coefficient of 0.03 as the input point. The *X*-directional response of the primary mirror component was analyzed, and the displacement frequency response curves in *X* directions are shown in Figure 10.

The results of the frequency response analysis show that the peak of the *X*-direction excitation occurs at 210 Hz, the displacement amplification is 8.75 times, the response amplification and the dynamic stress at the flexible groove were obtained, and the maximum stress is 101.4 MPa, which is well below the yield limit of the material selected. It indicates that the primary mirror assembly has good kinetic properties and no fracture damage occurs under kinetic loading.

## 6. Experimental Verification

Based on the optimized design results above, the primary mirror was cast into the primary mirror blank using the RB-SiC reaction sintering method, as shown in Figure 11, (a) is the front; (b) is the back being machined. The components, including the primary mirror, three flexible hinges, and back plate, were manufactured by precision mechanical and optical machining, and when the RMS value reached λ/50, the mirror was then vacuum coated. The film layer does not change the RMS value of the mirror and should be subjected to various environmental tests before vacuum coating [28]. The fabricated components are assembled by first connecting the flexible hinges to the primary mirror flange with screws dipped in epoxy glue and then connecting the flexible hinges to the primary mirror back plate with screws dipped in glue to prevent the screws from loosening during vibration testing or emission of the primary mirror assembly.

### 6.1. Surface Shape Error Testing of the Primary Mirror

High-precision interferometric measurement of the surface shape error at 632.8 nm wavelength, when the optical axis is horizontal, was performed with a Zygo interferometer with a room temperature of 22 ± 2 °C. The measurement principle is that the laser interferometer emits a plane wave, which is projected onto the aspherical primary mirror through the compensator, and the laser is reflected by the reflector and then returned to the interferometer again through the compensator to form interference fringes, which are then converted into a surface shape cloud map reflecting the mirror shape accuracy. The accuracy of the reflector surface shape accuracy test results is very important; the compensator laser interferometry is currently a more accurate method for the surface shape accuracy test. Generally, the testing of the primary mirror surface shape error is carried out before coating to prevent human factors from damaging the film layer during the testing process. The surface shape error test of the primary mirror before coating is shown in Figure 12, and the surface shape error test cloud of the primary mirror is shown in Figure 13. The test results show that the surface error of the primary mirror has a root-mean-square value of 0.02λ, which satisfied the design index requirements.

### 6.2. Mechanical Vibration Test of the Primary Mirror Assembly

A mechanical vibration test was conducted on the primary mirror assembly to measure the dynamic acceleration response amplification of the primary mirror assembly and to verify the stiffness and stability of the primary mirror assembly. The vibration test was a low-frequency sinusoidal sweep at 0.2 g. In order to exclude the influence of vibration transmission caused by the fixture, the sensor was mounted on the back plate end face of the primary mirror assembly as close as possible to the part connected by screws to obtain a feedback signal to measure the motion characteristics of the primary mirror assembly, which can be a single point or multiple points. After the sensor was installed, the wires were connected, and the wires were fixed to the assembly or vibration tooling with adhesive tape to avoid interference due to the relative motion of the wire head and the sensor. The test was performed in the X direction. The X-direction vibration results are shown in Figure 14. From the vibration test results, it can be seen that the fundamental frequency of the primary mirror assembly in the X direction is 208.25 Hz, which is consistent with the fundamental frequency of the simulation analysis results. The first-order resonant frequency of not less than 150 Hz, as required by the overall satellite design index value, is satisfied, indicating that the primary mirror assembly has sufficiently high stability and dynamic characteristics.

## 7. Conclusions

The purpose of this paper is to optimize the design of the primary mirror and components of the micro/nano satellite space camera to obtain high surface shape accuracy, high stability, and ultra-lightweight. The material selection of the primary mirror and the initial structure after process improvement are studied in terms of optimized design and simulation analysis. According to this mathematical model, a parametric optimization design of the improved initial structure was carried out. The final optimized structure was obtained with the minimum value of the compromise performance as the optimization objective, the structural parameters of the mirror and the flexible hinge as the variables, and the surface shape accuracy, first-order frequency, and weight as the restraints. As the space camera is tested, launched, and operated in orbit, the primary mirror and the back plate are connected by flexible hinges. To better simulate the actual working condition, the primary mirror, the flexible hinges, and the back plate of the primary mirror were analyzed in simulation and verified experimentally together. The results of finite element simulation analysis show that the RMS value of the mirror is less than the design index of 12.6 nm when subjected to gravity in X, Y, and Z directions, the maximum displacement of the primary mirror is 4.854 μm, which is less than the maximum 10 μm required by the design index, and the maximum inclination angle is 2.860″, which is less than the design index of 5″. After modal analysis, the first resonance frequency of the mirror assembly is 203.74 Hz. The resonance of the primary mirror assembly occurs at 210 Hz after frequency response sweep analysis, and the largest dynamic stress of 101.4 MPa occurs at the flexible hinge, which will not fracture. The quality of the primary mirror is 8.66 kg, which is less than the maximum limit value of 10 kg. Finally, the primary mirror assembly was precision machined, fabricated, and assembled. The surface error measurement of the primary mirror was tested by the ZYGO interferometer, and the test value was 0.02λ. The primary mirror assembly was tested by mechanical vibration test, the fundamental frequency was 208.25 Hz, and the test value was consistent with the simulation analysis results. The finite element analysis and test results indicate that all indexes of the primary mirror assembly meet the system’s general requirements. This study can provide an important reference value for the optimized design of a primary mirror of a space camera.

## Figures and Tables

**Figure 1 sensors-23-05441-f001:**
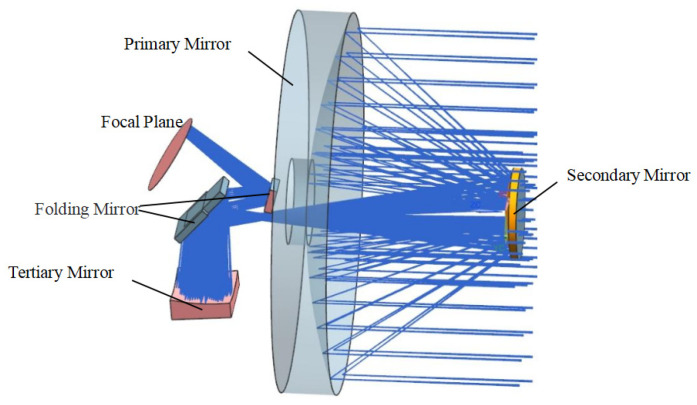
The coaxial triple reflection optical system layout.

**Figure 2 sensors-23-05441-f002:**
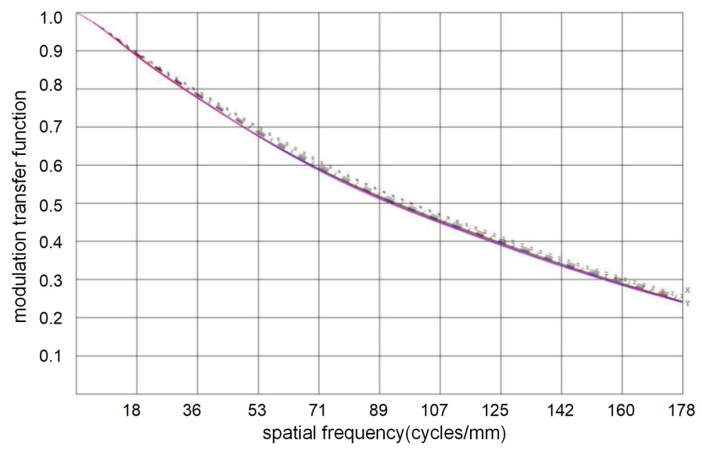
The MTF of the coaxial triple reflection optical system (Nyquist).

**Figure 3 sensors-23-05441-f003:**
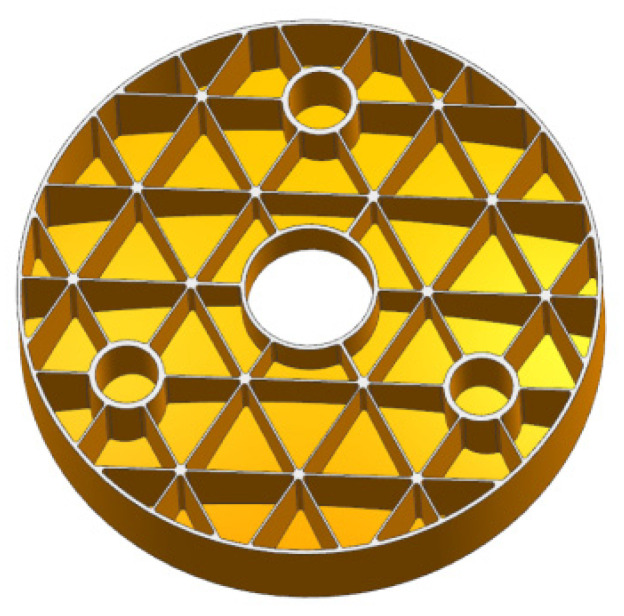
The initial structure of the primary mirror.

**Figure 4 sensors-23-05441-f004:**
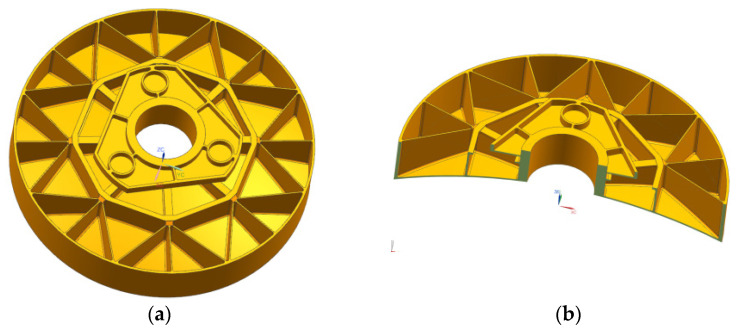
The initial structure of the primary mirror after a process improvement. (**a**) stereogram; (**b**) cutaway view.

**Figure 5 sensors-23-05441-f005:**
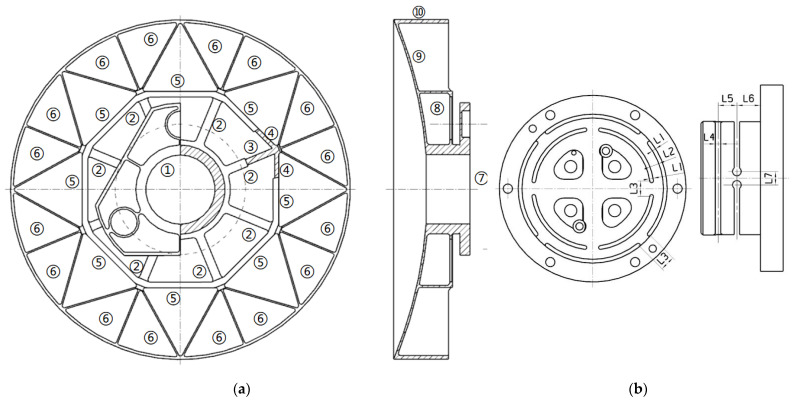
The optimization parameters of the primary mirror and the flexible hingle. (**a**) Primary mirror optimization parameters; (**b**) The flexure hinge optimization parameters.

**Figure 6 sensors-23-05441-f006:**
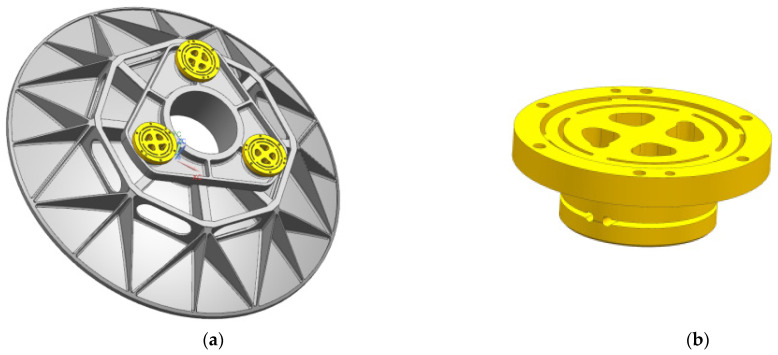
The optimized primary mirror assembly and flexible hinge structure. (**a**) The primary mirror assembly; (**b**) the flexure hinge.

**Figure 7 sensors-23-05441-f007:**
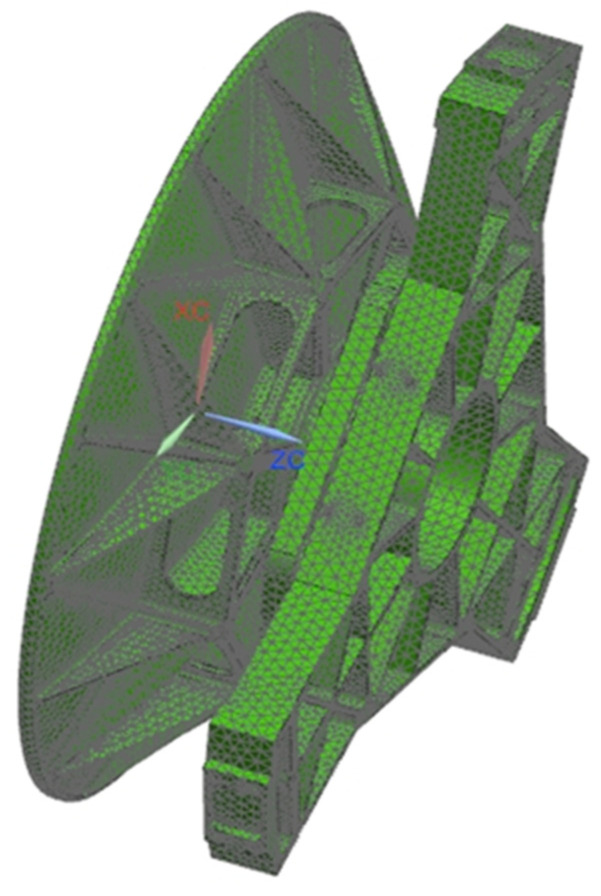
The finite element mesh model of the primary mirror assembly.

**Figure 8 sensors-23-05441-f008:**
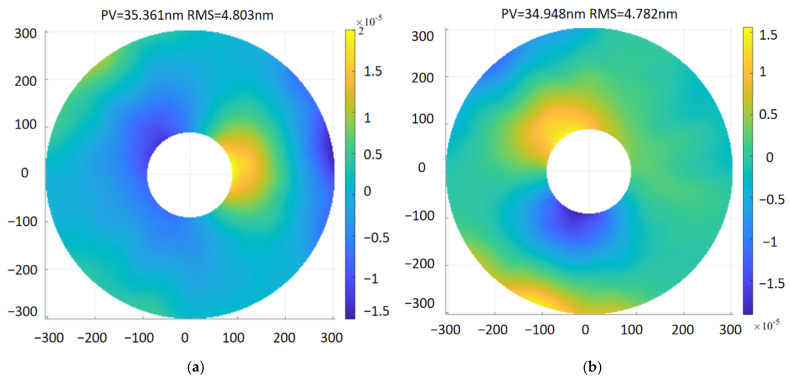
The optimized surface shape error cloud of the primary mirror under the six load cases. (**a**) X gravity; (**b**) Y gravity; (**c**) Z gravity; (**d**) 4 °C temperature rise; (**e**) 0.01 mm assembly error in Y; (**f**) 0.01 mm assembly error in Z.

**Figure 9 sensors-23-05441-f009:**
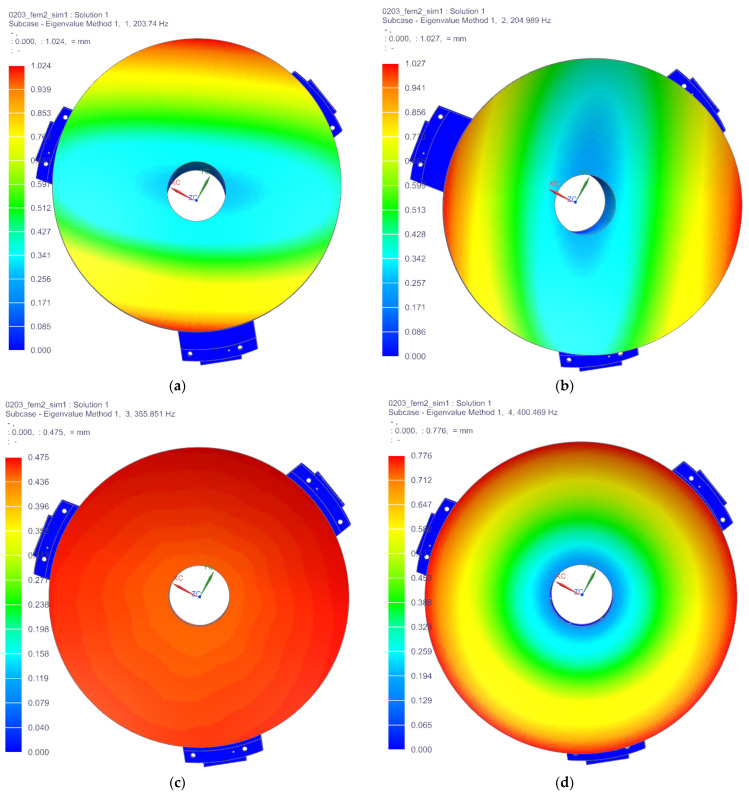
The results of the first six natural modes of vibration analysis. (**a**–**f**) The first to sixth vibration shape diagram.

**Figure 10 sensors-23-05441-f010:**
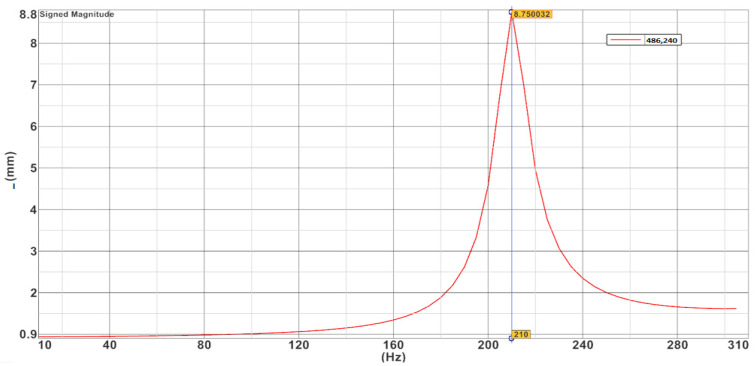
The displacement frequency response curves in *X* directions.

**Figure 11 sensors-23-05441-f011:**
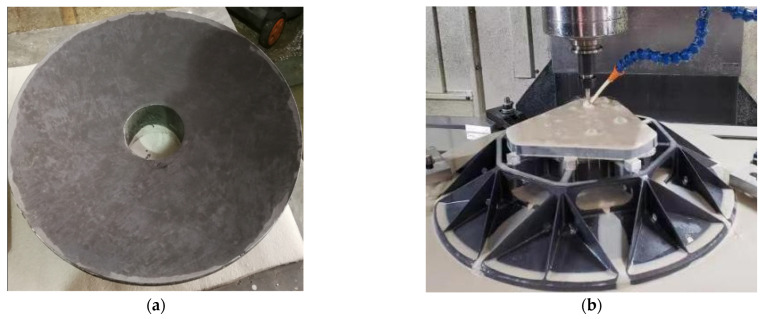
The primary mirror blank. (**a**) The front; (**b**) the back being machined.

**Figure 12 sensors-23-05441-f012:**
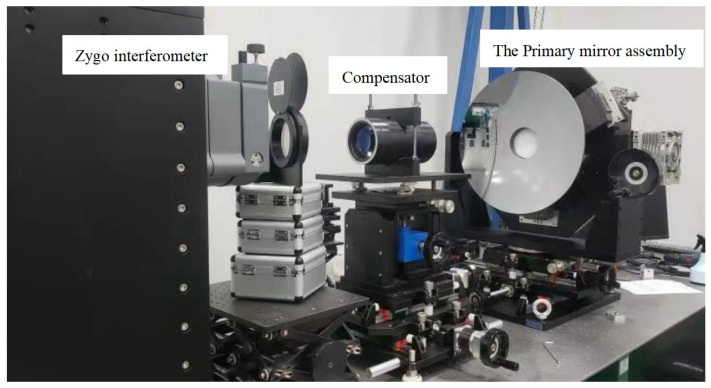
The surface shape error test of the primary mirror before coating.

**Figure 13 sensors-23-05441-f013:**
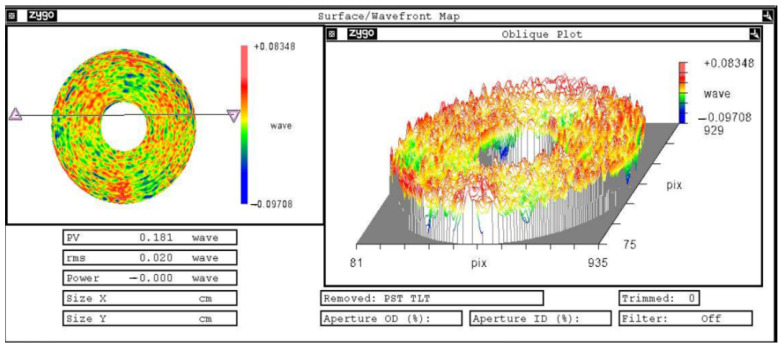
The surface shape error test cloud of the primary mirror.

**Figure 14 sensors-23-05441-f014:**
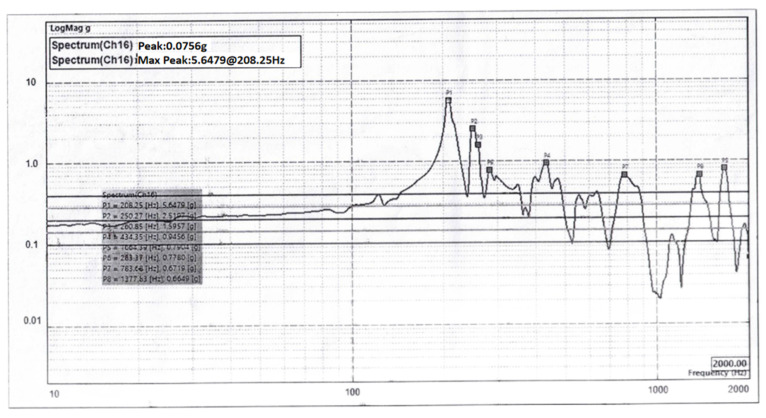
The X-direction vibration results.

**Table 1 sensors-23-05441-t001:** The detailed design index of the primary mirror assembly.

Components	Value
The Primary Mirror Mass	<10kg
RMS Error Under Gravity (X, Y, Z)	<λ/50 (12.6 nm, λ=632.8 nm)
RMS error under uniform temperature rise 4 °C	<λ/50 (12.6 nm)
RMS error under 0.01 mm planeness error	<λ/50 (12.6 nm)
PV error under gravity	<λ/10 (63.3 nm)
Fundamental frequency	>150 Hz
Operating temperature	20 ± 4 °C
Primary mirror around *X* or *Y* inclination angle	<5″
Linear Displacement of the Mirror Under 1 g Gravity	<10 μm

**Table 2 sensors-23-05441-t002:** Material characteristics of the primary mirror assembly.

Components	Material	Young’ Modulus (Gpa)	Poisson’ Ratio	Density (g/cm^3^)	Coefficient of Thermal Expansion (10^−6^/K)
Primary mirror	SiC	330	0.25	3.05	2.5
Flexure	4J32	141	0.28	8.1	2.5

**Table 3 sensors-23-05441-t003:** The original parameters and optimal results of the primary mirror and flexure hinge.

Variable	Domain (mm)	Original (mm)	Optimized (mm)
rib ①	8 ≤ (1) ≤ 10	8	10
rib ②	10 ≤ (2) ≤ 15	13	10
rib ③	2 ≤ (3) ≤ 5	3	4
rib ④	2 ≤ (4) ≤ 5	3	3
rib ⑤	6 ≤ (5) ≤ 10	10	8
rib ⑥	2 ≤ (6) ≤ 3	3	2
Diameter ⑦	Φ260 ≤ (7) ≤ Φ350	Φ350	Φ308
rib height ⑧	55 ≤ (8) ≤ 70	70	60
Thickness ⑨	3 ≤ (9) ≤ 4	3	3
rib height ⑩	3 ≤ (10) ≤ 70	5	4
L1	1 ≤ (L1) ≤ 3	2	1.5
L2	2 ≤ (L2) ≤ 6	6	3.5
L3	3 ≤ (L3) ≤ 6	3	4
L4	1 ≤ (L4) ≤ 3	3	1.5
L5	5 ≤ (L5) ≤ 10	8	6.5
L6	3 ≤ (L6) ≤ 5	5	3.5

**Table 4 sensors-23-05441-t004:** The finite element analysis results of the primary mirror assembly.

Loads	Surface Shape Error (nm)		Linear Displacement (um)			Inclination Angle (″)	
	PV	RMS	Tx	Ty	Tz	Rx	Ry
X gravity	35.361	4.803	0	0.094	0.01	0.177	0
Y gravity	34.948	4.782	0.094	0	0	0	0.178
Z gravity	28.458	5.941	0	0	0.176	0.001	0
4 °C temperature rise	3.246	0.553	1.007	1.670	0.935	0.033	0.036
0.01 mm assembly error in Y	40.480	6.729	1.709	0.930	1.837	1.796	2.290
0.01 mm assembly error in Z	35.59	3.24	0.011	1.572	4.854	2.860	0.024
Design index	63.3	12.6	10	10	10	5	5

**Table 5 sensors-23-05441-t005:** The first six frequencies and vibration modes of the primary mirror assemble.

Order	Frequency/Hz	Vibration Mode
1	203.74	Rotate along *X* axis
2	204.989	Rotate along *Y* axis
3	355.851	Move along *Z* axis
4	400.469	Rotate along *Z* axis
5	683.427	Move along *X* axis
6	684.067	Move along *Y* axis

## Data Availability

Not applicable.

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
