# Peer review of "Optimization Design of Large-Aperture Primary Mirror for a Space Remote Camera"

_sensors, 2023, doi:10.3390/s23125441_

Round 1
Reviewer 1 Report
The primary mirror is the core component of the high-resolution space remote sensing camera and its surface shape accuracy and structural stability determine the imaging quality of the optical system. Thus, the optimized design of primary mirror is very important. This study presents the whole optimized design process from the perspective of overall design and considers the manufacturing process, material selection, mathematical model of size optimization, connection with the flexible hinge and so on. This work is a good example of engineering optimization and is well written. Thus, it is commended for publication after some minor modifications.
1) The article does not explain what is a micro-nano satellite and what are its characteristics? What is the connection with the camera on board?
2) The abbreviation of RMS needs to be explained when it first appears in the abstract.
no
Reviewer 2 Report
This manuscript demonstrates the mechanical design of a large-aperture SiC mirror with a diameter of 610 mm. The flange and mirror body are integrated molded. The support force is applied to the flange through a flexible joint, reducing the sensitivity of surface deformation to shock, vibration, and temperature changes. The structural parameters of the mirror and the flexible hinge are optimized based on a constrained optimization method. The finite element simulation of the primary mirror assembly is carried out for different working conditions. Finally, the mirror assembly is built and tested, where the test results are consistent with the simulation results.
There are some details of the manuscript that require correction, which are noted below:
1) In Section 3.1, more details should be provided to explain the superiority of choosing SiC to other commonly used materials in reflective mirrors.
2) The Section ‘3.3.2’ should be modified as Section ‘3.2’.
3) In line 202, please declare why the mirror thickness is taken as 80mm.
4) In Eq. (8), please declare the meaning of the symbols ‘µ’ and ‘p’.
5) In Fig. 8, the surface deformation map of the mirror is not practical, since there is a hole in the center of the mirror.
6) In Section 5.1, please declare how the assembly error is applied in the finite element model.
7) In Section 6.2, please illustrate the setup of the sensors in the mechanical vibration test of the primary mirror assembly.
Reviewer 3 Report
This manuscript makes a complete design and analysis, including optical performance and mechanical property, on the primary mirror of a three mirror telescope optics. The result is also verified with a prototype and associated measurement. Although the work is quite complete and the aperture size is considered quite large, the structure of the whole optical system is a traditional one, and the design process also follows the standard procedure. It is suggested that the author could make a comparison with the existing primary mirror of the same kind to highlight the novelty or technical breakthrough contributed in the work.
There are quite a few typos in the manuscript which needs proof reading. The most obvious one is the title of section 7, which should be conclusions rather than convection.
Round 2
Reviewer 2 Report
The manuscript has been significantly improved. And, all my concerns have been addressed properly.